# Agreement between ranking metrics in network meta-analysis: an empirical study

Virginia Chiocchia ![ORCID],[1] Adriani Nikolakopoulou,[1] Theodoros Papakonstantinou,[1] Matthias Egger ![ORCID],[1,2] Georgia Salanti ![ORCID] [1]

[1]Institute of Social and Preventive Medicine, University of Bern, Bern, BE, Switzerland
[2]Population Health Sciences, Bristol Medical School, University of Bristol, Bristol, United Kingdom

**Correspondence to**
Virginia Chiocchia;
virginia.chiocchia@ispm.unibe.ch

## ABSTRACT

**Objective** To empirically explore the level of agreement of the treatment hierarchies from different ranking metrics in network meta-analysis (NMA) and to investigate how network characteristics influence the agreement.

**Design** Empirical evaluation from re-analysis of NMA.

**Data** 232 networks of four or more interventions from randomised controlled trials, published between 1999 and 2015.

**Methods** We calculated treatment hierarchies from several ranking metrics: relative treatment effects, probability of producing the best value $p_{(BV)}$ and the surface under the cumulative ranking curve (SUCRA). We estimated the level of agreement between the treatment hierarchies using different measures: Kendall's $\tau$ and Spearman's $\rho$ correlation; and the Yilmaz $\tau_{AP}$ and Average Overlap, to give more weight to the top of the rankings. Finally, we assessed how the amount of the information present in a network affects the agreement between treatment hierarchies, using the average variance, the relative range of variance and the total sample size over the number of interventions of a network.

**Results** Overall, the pairwise agreement was high for all treatment hierarchies obtained by the different ranking metrics. The highest agreement was observed between SUCRA and the relative treatment effect for both correlation and top-weighted measures whose medians were all equal to 1. The agreement between rankings decreased for networks with less precise estimates and the hierarchies obtained from $p_{BV}$ appeared to be the most sensitive to large differences in the variance estimates. However, such large differences were rare.

**Conclusions** Different ranking metrics address different treatment hierarchy problems, however they produced similar rankings in the published networks. Researchers reporting NMA results can use the ranking metric they prefer, unless there are imprecise estimates or large imbalances in the variance estimates. In this case treatment hierarchies based on both probabilistic and non-probabilistic ranking metrics should be presented.

## INTRODUCTION

Network meta-analysis (NMA) is being increasingly used by policymakers and clinicians to answer one of the key questions in medical decision-making: 'what treatment works best for the given condition?'.[1 2] The

### Strengths and limitations of this study

► To our knowledge, this is the first empirical study exploring the level of agreement of the treatment hierarchies from different ranking metrics in network meta-analysis (NMA).
► The study also explores how agreement is influenced by network characteristics.
► More than 200 published NMAs were re-analysed and three different ranking metrics calculated using both frequentist and Bayesian approaches.
► Other potential factors not investigated in this study could influence the agreement between hierarchies.

relative treatment effects, estimated in NMA, can be used to produce ranking metrics: statistical quantities measuring the performance of an intervention on the studied outcomes, thus producing a treatment hierarchy from the most preferable to the least preferable option.[3 4]

Despite the importance of treatment hierarchies in evidence-based decision-making, various methodological issues related to the ranking metrics have been contested.[5–7] This ongoing methodological debate focusses on the uncertainty and bias in a single ranking metric. Hierarchies produced by different ranking metrics are not expected to agree because ranking metrics differ. For example, a *non-probabilistic ranking metric* such as the treatment effect against a common comparator considers only the mean effect (eg, the point estimate of the odds ratio (OR)) and ignores the uncertainty with which this is estimated. In contrast, the probability that a treatment achieves a specific rank (a *probabilistic ranking metric*) considers the entire estimated distribution of each treatment effect. However, it is important to understand why and how rankings based on different metrics differ.

There are network characteristics that are expected to influence the agreement of

treatment hierarchies from different ranking metrics, such as the precision of the included studies and their distribution across treatment comparisons.[4 8] Larger imbalances in precision in the estimation of the treatment effects affects the agreement of the treatment hierarchies from probabilistic ranking metrics, but it is currently unknown whether in practice these imbalances occur and whether they should inform the choice between different ranking metrics. To our knowledge, no empirical studies have explored the level of agreement of treatment hierarchies obtained from different ranking metrics, or examined the network characteristics likely to influence the level of agreement. Here, we empirically evaluated the level of agreement between ranking metrics and examined how the agreement is affected by network features. The article first describes the methods for the calculation of ranking metrics and of specific measures to assess the agreement and to explore factors that affects it, respectively. Then, a network featuring one of the explored factors is shown as an illustrative example to display differences in treatment hierarchies from different ranking metrics. Finally, we present the results from the empirical evaluation and discuss their implications for researchers undertaking NMA.

## METHODS

### Data

We re-analysed networks of randomised controlled trials from a database of articles published between 1999 and 2015, including at least four treatments; details about the search strategy and inclusion/exclusion criteria can be found in.[9 10] We selected networks reporting arm-level data for binary or continuous outcomes. The database is accessible in the *nmadb* R package.[11]

### Re-analysis and calculation of ranking metrics

All networks were re-analysed using the relative treatment effect that the original publication used: OR, risk ratio (RR), standardised mean difference (SMD) or mean difference (MD). We estimated relative effects between treatments using a frequentist random-effects NMA model using the *netmeta* R package.[12] For the networks reporting ORs and SMDs we re-analysed them also using Bayesian models using self-programmed NMA routines in JAGS (https://github.com/esm-ispm-unibech/NMAJags). To obtain probabilistic ranking metrics in a frequentist setting, we used parametric bootstrap by producing 1000 data sets from the estimated relative effects and their variance-covariance matrix. By averaging over the number of simulated relative effects we derived the *probability of treatment i to produce the best value*

$$p_{i,BV} := p_{i,1} = P\left(\mu_{ij} > 0 \ \forall j \in \mathbb{T}\right)$$

where $\mu_{ij}$ is the estimated mean relative effect of treatment $i$ against treatment $j$ out of a set $\mathbb{T}$ of $T$ competing treatments. We will refer to this as $p_{BV}$. This ranking metric indicates how likely a treatment is to produce the largest

values for an outcome (or smallest value, if the outcome is harmful). We also calculated the surface under the cumulative ranking curve ($SUCRA^F$)[3]

$$SUCRA_i = \frac{\sum_{r=1}^{T-1} c_{i,r}}{T-1}$$

where $c_{i,r} = \sum_{v=1}^{r} p_{i,v}$ are the cumulative probabilities that treatment $i$ will produce an outcome that is among the $r$ best values (or that it outperforms $T - r$ treatments). SUCRA, unlike $p_{BV}$, also considers the probability of a treatment to produce unfavourable outcome values. Therefore, the treatment with the largest SUCRA value represents the one that outperforms the competing treatments in the network, meaning that overall it produces preferable outcomes compared with the others. We also obtained SUCRAs within a Bayesian framework ($SUCRA^B$).

To obtain the non-probabilistic ranking metric we fitted an NMA model and estimated related treatment effects. To obtain estimates for all treatments we reparametrise the NMA model so that each treatment is compared with a fictional treatment of average performance.[13 14] The estimated relative effects against a fictional treatment $F$ of average efficacy $\hat{\mu}_{iF}$ represent the ranking metric and the corresponding hierarchy is obtained simply by ordering the effects from the largest to the smallest (or in ascending order, if the outcome is harmful). The resulting hierarchy is identical to that obtained using relative effects from the conventional NMA model, irrespective of the reference treatment. In the rest of the manuscript, we will refer to this ranking metric simply as relative treatment effect.

### Agreement between ranking metrics

To estimate the level of agreement between the treatment hierarchies obtained using the three chosen ranking methods we employed several correlation and similarity measures.

To assess the correlation between ranking metrics we used Kendall's $\tau$[15] and the Spearman's $\rho$.[16] Both Kendall's $\tau$ and Spearman's $\rho$ give the same weight to each item in the ranking. In the context of treatment ranking, the top of the ranking is more important than the bottom. We therefore also used a top-weighted variant of Kendall's $\tau$, Yilmaz $\tau_{AP}$,[17] which is based on a probabilistic interpretation of the average precision measure used in information retrieval[18] (see online supplementary appendix).

The measures described so far can only be considered for conjoint rankings, that is, for lists where each item in one list is also present in the other list. Rankings are *non-conjoint* when a ranking is truncated to a certain *depth k* with such lists called *top-k rankings*. We calculated the Average Overlap,[19 20] a top-weighted measure for top-k rankings that considers the cumulative intersection (or *overlap*) between the two lists and averages it over a specified depth (cut-off point) $k$ (see online supplementary appendix for details). We calculated the Average Overlap between pairs of rankings for networks with at least six

treatments (139 networks) for a depth $k$ equal to half the number of treatments in the network, $k = \frac{T}{2}$ (or $(T-1)/2$ if $T$ is an odd number).

We calculated the four measures described above to assess the pairwise agreement between the three ranking metrics within the frequentist setting and summarised them for each pair of ranking metrics and each agreement measure using the median and the first and third quartiles. The hierarchy according to $SUCRA^B$ was compared with that of its frequentist equivalent to check how often the two disagree.

### Influence of network features on the rankings agreement

The main network characteristic considered was the amount of information in the network (reflected in the precision of the estimates). Therefore, for each network we calculated the following measures of information:

► The average variance, calculated as the mean of the variances of the estimated treatment effects $mean\left(SE^2\right)$, to show how much information is present in a network altogether;

► The relative range of variance, calculated as $\frac{\max SE^2 - \min SE^2}{\max SE^2}$, to describe differences in information about each intervention within the same networks;

► The total sample size of a network over the number of interventions.

These measures are presented in scatter plots against the agreement measurements for pairs of ranking metrics.

All the codes for the empirical evaluation are available at https://github.com/esm-ispm-unibe-ch/rankingagreement

### Patient and public involvement

Patients and the public were not involved in this study.

### ILLUSTRATIVE EXAMPLE

To illustrate the impact of the amount of information on the treatment hierarchies from different ranking metrics, we used a network of nine antihypertensive treatments for primary prevention of cardiovascular disease that presents large differences in the precision of the estimates of overall mortality.[21] The network graph and forest plot of relative treatment effects of each treatment versus placebo are presented in figure 1. The relative treatment effects reported are RR estimated using a random effects NMA model.

Table 1 shows the treatment hierarchies obtained using the three ranking metrics described above. The highest overall agreement is between hierarchies from the $SUCRA^F$ and the relative treatment effect as shown by both correlation (Spearman's $\rho = 0.93$, Kendall's $\tau = 0.87$) and top-weighted measures (Yilmaz's $\tau_{AP} = 0.87$; Average Overlap=0.85). The level of agreement decreases when $SUCRA^F$ and the relative treatment effect are compared with $p_{BV}$ rankings (Spearman's $\rho = 0.63$ and $\rho = 0.85$, respectively). Agreement with $p_{BV}$ especially decreases when considering top ranks only (Average Overlap is 0.48 for $p_{BV}$ vs $SUCRA^F$ and 0.54 for $p_{BV}$ vs relative treatment effect). All agreement measures are presented in online supplementary table S1.

The reason for this disagreement is explained by the differences in precision in the estimated effects (figure 1). These RRs versus placebo range from 0.82 (diuretic/beta-blocker vs placebo) to 0.98 (beta-blocker vs placebo). All estimates are fairly precise except for the RR of conventional therapy versus placebo whose 95% confidence interval (CI) extends from 0.21 to 3.44. This uncertainty in the estimation is due to the fact that conventional therapy is compared only with angiotensin receptor blockers (ARB) via a single study. This large difference in the precision of the estimation of the treatment effects mostly affects the $p_{BV}$ ranking, which disagrees the most with both of the other rankings. Consequently, the conventional therapy is in the first rank in the $p_{BV}$ hierarchy (because of the large uncertainty) but only features in the third/fourth and sixth rank using the relative treatment effects and $SUCRA^F$ hierarchies, respectively.

To explore how the hierarchies for this network would change in case of increased precision, we reduced the standard error (SE) of the conventional versus ARB

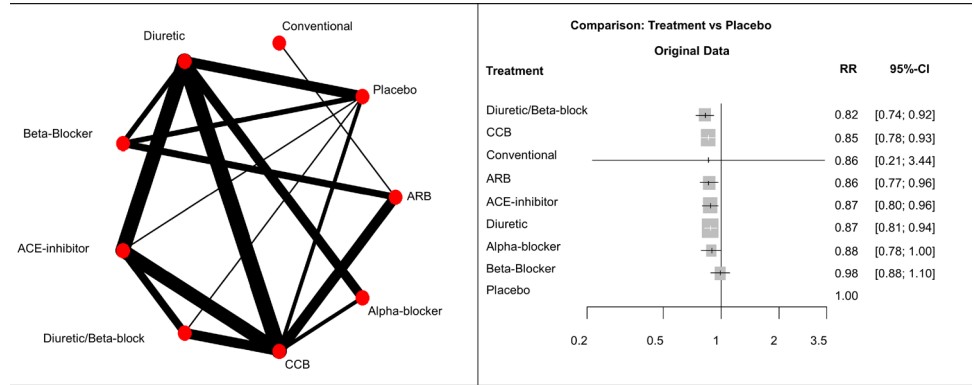

**Figure 1** (Left panel) Network graph of network of nine antihypertensive treatments for primary prevention of cardiovascular disease. Line width is proportional to inverse standard error of random effects model comparing two treatments. (Right panel) Forest plots of relative treatment effects of overall mortality for each treatment versus placebo. ACE, Angiotensin Converting Enzyme; ARB, angiotensin receptor blockers; CCB, calcium channelblockers; RR, risk ratio.

**Table 1** Example of treatment hierarchies from different ranking metrics for a network of nine antihypertensive treatment for primary prevention of cardiovascular disease

| Treatment | Original data | | | Fictional data with increased precision for conventional treatment versus ARB | | |
| | $p_{BV}$ Ranks | $SUCRA_F$ Ranks | Relative treatment effect ranks | $p_{BV}$ Ranks | $SUCRA_F$ Ranks | Relative treatment effect ranks |
|---|---|---|---|---|---|---|
| Conventional | 1 | 6 | 3.5 | 3 | 4 | 3.5 |
| Diuretic/beta-blocker | 2 | 1 | 1 | 1 | 1 | 1 |
| ARB | 3 | 3 | 3.5 | 4.5 | 3 | 3.5 |
| CCB | 4 | 2 | 2 | 2 | 2 | 2 |
| Alpha-blocker | 5 | 7 | 7 | 4.5 | 7 | 7 |
| ACE-inhibitor | 6 | 4 | 5 | 6.5 | 5 | 5 |
| Diuretic | 7 | 5 | 6 | 6.5 | 6 | 6 |
| Placebo | 8.5 | 9 | 9 | 8.5 | 9 | 9 |
| Beta-blocker | 8.5 | 8 | 8 | 8.5 | 8 | 8 |

Relative treatment effect stands for the relative treatment effect against fictional treatment of average performance. The first three rankings from the left-hand side are obtained using the original data; the equivalent three rankings on the right-hand side are produced by reducing the standard error of the conventional versus ARB treatment effect from 0.7 to a fictional value of 0.01.
ACE, angiotensin converting enzyme; ARB, angiotensin receptor blockers; CCB, calcium channel blockers; $pBV$
, probability of producing the best value; SUCRA$_F$, surface under the cumulative ranking curve (calculated in frequentist setting).

treatment effect from the original 0.7 to a fictional value of 0.01 resulting in a CI 0.77 to 0.96. The columns in the right-hand side of table 1 display the three equivalent rankings after the SE reduction. The conventional treatment has moved up in the hierarchy according to $SUCRA^F$ and moved down in the one based on $p_{BV}$, as expected. The treatment hierarchies obtained from the $SUCRA^F$ and the relative treatment effect are now identical (conventional and ARB share the 3.5 rank because they have the same effect estimate) and the agreement with the $p_{BV}$ rankings also improved ($p_{BV}$ vs $SUCRA^F$ Spearman's $\rho = 0.89$, Average Overlap=0.85; $p_{BV}$ vs relative treatment effect Spearman's $\rho = 0.91$, Average Overlap=0.94; online supplementary table S1).

## RESULTS
A total of 232 networks were included in our data set. Their characteristics are shown in table 2.

The majority of networks (133 NMAs, 57.3%) did not report any ranking metrics in the original publication. Among those which used a ranking metric to produce a treatment hierarchy, the probability of being the best was the most popular metric followed by the SUCRA with 35.8% and 6.9% of networks reporting them, respectively.

Table 3 presents the medians and quartiles for each similarity measures. All hierarchies showed a high level of pairwise agreement, although the hierarchies obtained from the $SUCRA^F$ and the relative treatment effect presented the highest values for both unweighted and with top-weighted measures (all measures' median equals 1). Only four networks (less than 2%) had a Spearman's correlation between $SUCRA^F$ and the relative treatment effect less than 90% (not reported). The correlation

becomes less between the $p_{BV}$ rankings and those obtained from the other two ranking metrics with Spearman's $\rho$ median decreasing to 0.9 and Kendall's $\tau$ decreasing to 0.8. The Spearman's correlation between these rankings was less than 90% in about 50% of the networks (in 116 and 111 networks for $p_{BV}$ vs $SUCRA^F$ and $p_{BV}$ vs relative effect, respectively; results not reported). The pairwise agreement between the $p_{BV}$ rankings and the other rankings also decreased when considering only top ranks ($p_{BV}$ vs $SUCRA^F$ Yilmaz's $\tau_{AP} = 0.77$, Average Overlap=0.83; $p_{BV}$ vs relative treatment effect Yilmaz's $\tau_{AP} = 0.79$, Average Overlap=0.88).

The SUCRAs from frequentist and Bayesian settings ($SUCRA^F$ and $SUCRA^B$) were compared in 126 networks (82 networks using the Average Overlap measure) as these reported OR and SMD as original measures. The relevant rankings do not differ much as shown by the median values of the agreement measures all equal to 1 and their narrow interquartile ranges (IQRs) (table 3). Nevertheless, a few networks showed a much lower agreement between the two SUCRAs. These networks provide posterior effect estimates for which the normal approximation is not optimal, some of which due to rare outcomes. Such cases were however uncommon as in only 6% of the networks the Spearman's correlation between and was less than 90%. Plots for the normal distributions from the frequentist setting and the posterior distributions of the log ORs for a network with a Spearman's of 0.6 between the two SUCRAs is available in online supplementary figure S1.[22]

Figure 2 presents how Spearman's $\rho$ and the Average Overlap vary with the average variance of the relative

**Table 2** Characteristics of the 232 NMAs included in the re-analysis

| Characteristics of networks | Median | IQR |
| --- | --- | --- |
| Median number of treatments compared | 6 | (5 to 9) |
| Median number of studies included | 19 | (12 to 34) |
| Median total sample size | 6100 | (2514 to 17264) |
| | **Number of NMAs** | **%** |
| Beneficial outcome | 97 | 41.8% |
| Dichotomous outcome | 185 | 79.7% |
| Continuous outcome | 47 | 20.3% |
| Published before 2010 | 42 | 18.1% |
| Ranking metric used in original publication (non-exclusive) | | |
| Probability of producing the best value | 83 | 35.8% |
| Rankograms | 7 | 3% |
| Median or mean rank | 3 | 1.3% |
| SUCRA | 16 | 6.9% |
| Other | 2 | 0.9% |
| None | 133 | 57.3% |
| Published in general medicine journals* | 125 | 53.9% |
| Published in health services research journals† | 3 | 1.3% |
| Published in specialty journals | 104 | 44.8% |

*Includes the categories Medicine, General and Internal, Pharmacology and Pharmacy, Research and Experimental, Primary Healthcare.
†Includes the categories Healthcare Sciences and Services, Health Policy and Services.
IQR, interquartile range; NMA, network meta-analysis; SUCRA, surface under the cumulative ranking curve.

treatment effect estimates in a network (scatter plots for the Kendall's $\tau$ and the Yilmaz's $\tau_{AP}$ are available in online supplementary figure S2). The treatment hierarchies agree more in networks with more precise estimates (left hand side of the plots).

The association between Spearman's $\rho$ or Average Overlap and the relative range of variance in a network (here transformed to a double logarithm of the inverse values) are displayed in figure 3. On the right-hand side of each plot we can find networks with smaller differences in the precision of the treatment effect estimates. Treatment hierarchies for these networks show a larger agreement than for those with larger differences in precision. The plots of the impact of the relative range of variance on all measures are available in online supplementary figure S3.

The total sample size in a network over the number of interventions has a similar impact on the level of agreement between hierarchies. This confirms that the agreement between hierarchies increases for networks with a large total sample size compared with the number of treatments and, more generally, it increases with the amount of information present in a network (online supplementary figure S4).

## DISCUSSION

Our empirical evaluation showed that in practice the level of agreement between treatment hierarchies is overall high for all ranking metrics used. The agreement between treatment hierarchies from *SUCRA* and relative treatment effect was very often perfect. The agreement between the rankings from *SUCRA* or relative treatment effect and the ranking from $p_{BV}$ was good but decreased when the top-ranked interventions are of interest. The agreement is higher for networks with precise estimates and small imbalances in precision.

Simulation studies[6 23] using theoretical examples have shown the importance of accounting for the precision in the estimation of the treatment effects when a hierarchy is to be obtained. However, we show that cases of extreme imbalance in the precision of the treatment effects are rather uncommon.

**Table 3** Pairwise agreement between treatment hierarchies obtained from the different ranking metrics measured by Spearman $\rho$, Kendall $\tau$, Yilmaz $\tau_{AP}$ and Average Overlap

| | $p_{BV}$ vs $SUCRA_F$ | $SUCRA_F$ versus relative treatment effect | $p_{BV}$ versus relative treatment effect | $SUCRA_F$ versus $SUCRA_B$ |
| --- | --- | --- | --- | --- |
| Spearman $\rho$ | 0.9 (0.8 to 0.96) | 1 (0.99 to 1) | 0.9 (0.8 to 0.97) | 1 (0.98 to 1) |
| Kendall $\tau$ | 0.8 (0.67 to 0.91) | 1 (0.95 to 1) | 0.8 (0.69 to 0.91) | 1 (0.93 to 1) |
| Yilmaz $\tau_{AP}$ | 0.78 (0.6 to 0.9) | 1 (0.93 to 1) | 0.79 (0.65 to 0.9) | 1 (0.93 to 1) |
| Average Overlap | 0.85 (0.72 to 0.96) | 1 (0.91 to 1) | 0.88 (0.79 to 1) | 1 (0.94 to 1) |

Medians, first and third quartiles are reported.
Relative treatment effect stands for the relative treatment effect against fictional treatment of average performance.
$P_{BV}$, probability of producing the best value; $SUCRA_B$, surface under the cumulative ranking curve (calculated in Bayesian setting); $SUCRA_F$, surface under the cumulative ranking curve (calculated in frequentist setting).

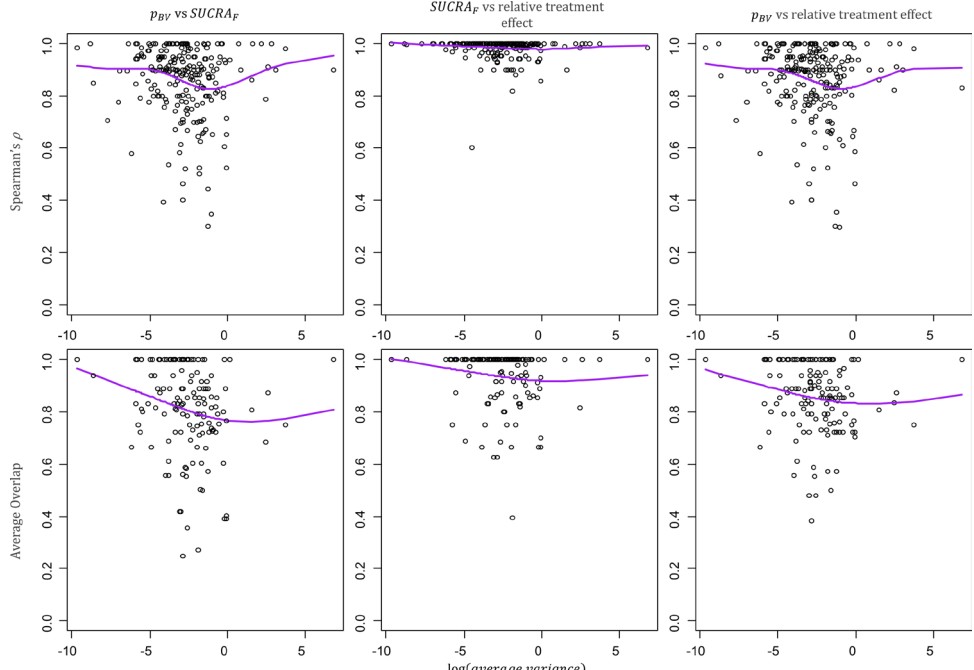

**Figure 2** Scatter plots of the average variance in a network and the pairwise agreement between hierarchies from different ranking metrics. The average variance is calculated as the mean of the variances of the estimated treatment effects and describes the average information present in a network. More imprecise network are on the right-hand side of the plots. Spearman $\rho$ (top row) and Average Overlap (bottom row) values for the pairwise agreement between $p_{BV}$ and $SUCRA_F$ (first column), $SUCRA_F$ and relative treatment effect (second column), $p_{BV}$ and relative treatment effect (third column). Purple line: cubic smoothing spline with five degrees of freedom. $p_{BV}$, probability of producing the best value; $SUCRA_F$, surface under the cumulative ranking curve (calculated in frequentist setting).

Several factors can be responsible for imprecision in the estimation of the relative treatment effects in a network:

► Large sampling error, determined by a small sample size, small number of events or a large standard deviation;

► Poor connectivity of the network, when only a few links and few closed loops of evidence connect the treatments;

► Residual inconsistency;

► Heterogeneity in the relative treatment effects.

Random-effects models tend to provide relative treatment effects with similar precision as heterogeneity increases. In contrast, in the absence of heterogeneity when fixed-effects models are used, the precision of the effects can vary a lot according to the amount of data available for each intervention. In the latter case, the ranking metrics are likely to disagree. Also, the role of precision in ranking disagreement is more pronounced in cases where the interventions have similar effects.

Our results also confirm that a treatment hierarchy can differ when the uncertainty in the estimation is incorporated into the ranking metric (by using, for example, a probabilistic metric rather than ranking the point estimate of the mean treatment effect)[8 24] and that rankings from the $p_{BV}$ seem to be the most sensitive to differences in precision in the estimation of treatment effects. We showed graphically that the agreement is less in networks with more uncertainty and with larger imbalances in the variance estimates. However, we also found that such large imbalances do not occur frequently in real data and in the majority of cases the different treatment hierarchies have a relatively high agreement.

We acknowledge that there could be other factors influencing the agreement between hierarchies that we did not explore, such as the chosen effect measures.[25] However, we think it is unlikely that such features play a big role in ranking agreement unless assumptions are violated or data in the network is sparse.[26] Adjustment via network meta-regression (for example, for risk of bias or small-study effects) might impact on the ranking of treatments not only by changing the point estimate but also by altering the total precision and the imbalance in the precision of the estimated treatment effects. We did not investigate the agreement between treatment hierarchies obtained from such adjusted analyses. We also did not explore non-methodological characteristics for networks with larger disagreement but we believe these characteristics are a proxy for the amount of information in a network, which is the main factor affecting the agreement between ranking metrics. For example, in some specific fields there are few or small randomised trials (eg, surgery) and, as a consequence, the resulting networks will have less information. Also, smaller (hence more imprecise) networks might be published more often in journal with lower impact factor and get less citations than large and precise networks.

To our knowledge, this is the first empirical study assessing the level of agreement between treatment

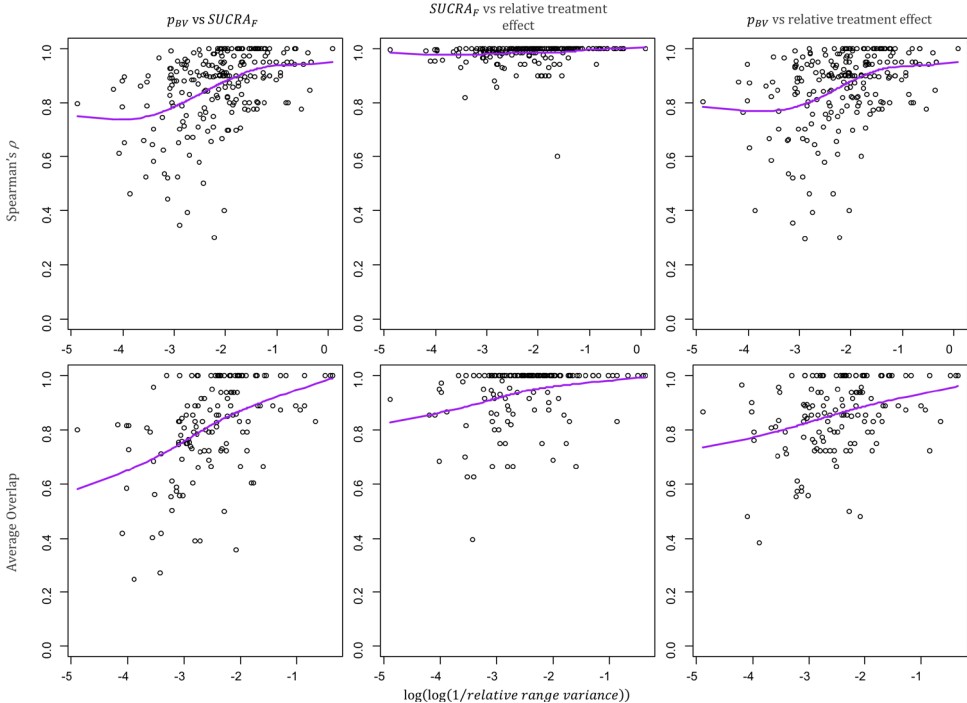

**Figure 3** Scatter plots of the relative range of variance in a network and the pairwise agreement between hierarchies from different ranking metrics. The relative range of variance, calculated as $\frac{maxSE^2 - minSE^2}{maxSE^2}$, indicates how much the information differs between interventions in the same networks. Networks with larger differences in variance are on the left-hand side of the plots. Spearman $\rho$ (top row) and Average Overlap (bottom row) values for the pairwise agreement between $p_{BV}$ and $SUCRA_F$ (first column), $SUCRA_F$ and relative treatment effect (second column), $p_{BV}$ and relative treatment effect (third column). Purple line: cubic smoothing spline with five degrees of freedom. $p_{BV}$, probability of producing the best value; ; $SUCRA_P$ surface under the cumulative ranking curve (calculated in frequentist setting).

hierarchies from ranking metrics in NMA and it provides further insights into the properties of the different methods. In this context, it is important to stress that neither the objective nor the findings of this empirical evaluation imply that a hierarchy for a particular metric works better or is more accurate than one obtained from another ranking metric. The reason why this sort of comparison cannot be made is that each ranking metric address a specific treatment hierarchy problem. For example, the *SUCRA* ranking addresses the issue of which treatment outperforms most of the competing interventions, while the ranking based on the relative treatment effect gives an answer to the problem of which treatment is associated with the largest average effect for the outcome considered.

Our study shows that, despite theoretical differences between ranking metrics and some extreme examples, they produce very similar treatment hierarchies in published networks. In networks with large amount of data for each treatment, hierarchies based on SUCRA or the relative treatment effect will almost always agree. Large imbalances in the precision of the treatment effect estimates do not occur often enough to motivate a choice between the different ranking metrics. Therefore, our advice to researchers presenting results from NMA is the following: *if the NMA estimated effects are precise,* to use the ranking metric they prefer; *if at least one NMA estimated effect is imprecise,* to refrain from making bold statements about treatment hierarchy and present hierarchies from both probabilistic (eg, SUCRA or rank probabilities) and non-probabilistic metrics (eg, relative treatments effects).

**Contributors** VC designed the study, analysed the data, interpreted the results of the empirical evaluation and drafted the manuscript. GS designed the study, interpreted the results of the empirical evaluation and revised the manuscript. AN provided input into the study design and the data analysis, interpreted the results of the empirical evaluation and revised the manuscript. TP developed and manages the database where networks' data was accessed, provided input into the data analysis and revised the manuscript. ME provided input into the study design and revised the manuscript. All the authors approved the final version of the submitted manuscript.

**Funding** This work was supported by the Swiss National Science Foundation grant/award number 179158.

**Competing interests** None declared.

**Patient consent for publication** Not required.

**Provenance and peer review** Not commissioned; externally peer reviewed.

**Data availability statement** Data are available in a public, open access repository. The data for the network meta-analyses included in this study are available in the database accessible using the nmadb R package.

**ORCID iDs**

Virginia Chiocchia http://orcid.org/0000-0002-6196-3308
Matthias Egger http://orcid.org/0000-0001-7462-5132
Georgia Salanti http://orcid.org/0000-0002-3830-8508

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
