## [Reviewer comments · BMJ Open]

ARTICLE DETAILS

TITLE (PROVISIONAL)	Agreement between ranking metrics in network meta-analysis: an empirical study
AUTHORS	Chiocchia, Virginia; Nikolakopoulou, Adriani; Papakonstantinou, Theodoros; Egger, Matthias; Salanti, Georgia

VERSION 1 - REVIEW

REVIEWER	Sandeep Tripathi University of Illinois College of Medicine Peoria, IL
REVIEW RETURNED	25-Feb-2020

GENERAL COMMENTS	Chiocchia and colleagues calculated the treatment hierarchies from several ranking metrics: relative treatment effects, probability of producing the best value and the surface under the cumulative ranking curve. They then estimated the level of agreement between the treatment hierarchies using correlation coefficients and other methods. They also assessed how the amount of the information present in a network affects the agreement between treatment hierarchies. I have found the paper to be very well written and clearly explains the premise methodology as well as discussion of the conclusions. This is however a statistically heavy manuscript and I am not qualified to comment on the novelty of the findings. This however manuscript will be of interest to investigators who have been involved in meta-analysis and comparative efficacy research
---

REVIEWER	Xiangyu Meng Team of Molecular Oncology, Institut Curie - UMR 144 - CNRS, Paris, France
REVIEW RETURNED	18-Apr-2020

GENERAL COMMENTS	Chiocchia and colleagues investigated the agreement among various NMA ranking metrics using empirical NMA data. They found a globally similar rankings produced with different metrics approaches, and they reported imprecision in effect estimates and imbalanced variance estimates could bring disagreement between the rankings. The manuscript seems complete to answer the question from a pure methodological perspective; however, it still could be interesting if the authors could further examine if any
--

	characterization could be made for the networks with a larger disagreement among the ranking metrics, from a non-methodological perspective. For example, are they enriched in studies from a specific field? Are they systematically biased for publication time or journal quality/impact?
--	---

REVIEWER	Anna Chaimani Inserm, Université de Paris, France I am a former PhD student of G. Salanti
REVIEW RETURNED	12-Jun-2020

GENERAL COMMENTS	The present paper aims to empirically investigate how often and under which conditions the three most commonly used ranking metrics agree, in terms of the treatment hierarchies they produce, when applied to the same network meta-analysis (NMA). The authors re-analyzed 232 networks of interventions with at least four interventions and assessed the agreement across the ranking metrics using different approaches. I find the article an important contribution to the field and a step forward to resolve the large debate on the reliability and the usefulness of ranking in NMA. Below I provide some comments for the authors to consider: Major comments  1. I believe it would be of interest to briefly compare the findings of this empirical study with those from simulation studies on treatment ranking. Although, existing simulation studies may have not compared different ranking metrics, some conclusions seem to be similar. For example, the problematic situation of large variation in the precision of treatment effect estimates has been stretched out before (see Davies and Galla. Degree irregularity and rank probability bias in network meta-analysis, arXiv:2003.07662). 2. Regarding the non-probabilistic ranking metric, the authors used a re-parametrization of the NMA model from which “the resulting hierarchy is identical to that obtained using relative effects from the conventional NMA model.” I think the authors should clarify which relative effects they refer to here. For networks with a clear control intervention, it seems straightforward. But for networks without a control intervention changing the reference treatment may yield different hierarchies. 3. I think the illustrative example fits better after the empirical evidence. 4. It seems to me that, when investigating the influence of study features, precision of treatment effects should also be considered in relation to their magnitude. See the fictional example below where the precision for all three comparisons is the same in the two situations (seBA=.3061861, seCA=.3061862, seCB=.3535533) and there is full agreement in hierarchy between relative effects (against A), SUCRA and P(best). A moderate increase in the precision of comparison CB would not affect the agreement of the right hand side ranking, whereas the same increase would affect the agreement of the
--

	ranking at the left.  Minor comments  1. The authors report that “a few networks showed a much lower agreement between the two SUCRAs (page 12). These networks provide posterior effect estimates for which the Normal approximation is not optimal.” Did these networks have some specific characteristics? 2. Page 14, “Our results also confirm that a treatment hierarchy can differ when the uncertainty in the estimation is incorporated into the ranking metric”: I am not sure I understand what the authors imply here. 3. Page 14, “We acknowledge that there could be other factors influencing the agreement between hierarchies that we did not explore, such as the risk of bias...”. I cannot see how risk of bias could influence the agreement of the ranking since none of the metrics investigated takes this aspect into account.
--	--

VERSION 1 – AUTHOR RESPONSE

Reviewer(s)' Comments to Author:

Reviewer: 1
 Reviewer Name
 Sandeep Tripathi

Institution and Country
 University of Illinois College of Medicine Peoria, IL

Please state any competing interests or state 'None declared':
 None declared

Please leave your comments for the authors below Chiochia and colleagues calculated the treatment hierarchies from several ranking metrics: relative treatment effects, probability of producing the best value and the surface under the cumulative ranking curve. They then estimated the level of agreement between the treatment hierarchies using correlation coefficients and other methods. They also assessed how the amount of the information present in a network affects the agreement between treatment hierarchies.

I have found the paper to be very well written and clearly explains the premise methodology as well as discussion of the conclusions. This is however a statistically heavy manuscript and I am not qualified to comment on the novelty of the findings. This however manuscript will be of interest to investigators who have been involved in meta-analysis and comparative efficacy research

We thank the reviewer for their comments.

Reviewer: 2

Reviewer Name

Xiangyu Meng

Institution and Country

Team of Molecular Oncology, Institut Curie - UMR 144 - CNRS, Paris, France

Please state any competing interests or state 'None declared':

None

Please leave your comments for the authors below Chiocchia and colleagues investigated the agreement among various NMA ranking metrics using empirical NMA data. They found a globally similar rankings produced with different metrics approaches, and they reported imprecision in effect estimates and imbalanced variance estimates could bring disagreement between the rankings. The manuscript seems complete to answer the question from a pure methodological perspective; however, it still could be interesting if the authors could further examine if any characterization could be made for the networks with a larger disagreement among the ranking metrics, from a non-methodological perspective. For example, are they enriched in studies from a specific field? Are they systematically biased for publication time or journal quality/impact?

We thank the reviewer for their comments. There were only a few networks with large disagreement and this does not allow us to sensibly generate hypotheses about their characteristics. Any non-methodological characteristic must be associated with the size of the network, (e.g. smaller, therefore more imprecise, networks are generally published in journals with lower impact factors). We added the following statement in the Discussion section on page 15: "We also did not explore non-methodological characteristics for networks with larger disagreement but we believe these characteristics are a proxy for the amount of information in a network, which is the main factor affecting the agreement between ranking metrics. For example, in some specific fields there are few or small RCTs (e.g. surgery) and, as a consequence, the resulting networks will have less information. Also, smaller (hence more imprecise) networks might be published more often in journal with lower impact factor and get less citations than large and precise networks."

Reviewer: 3

Reviewer Name

Anna Chaimani

Institution and Country

Inserm, Université de Paris, France

Please state any competing interests or state 'None declared':

I am a former PhD student of G. Salanti

Please leave your comments for the authors below My comments are available in the attached document.

We thank the reviewer for her very helpful and detailed comments. We have addressed them specifically and updated the manuscript accordingly.

Major comments

I believe it would be of interest to briefly compare the findings of this empirical study with those from simulation studies on treatment ranking. Although, existing simulation studies may have not compared different ranking metrics, some conclusions seem to be similar. For example, the problematic situation of large variation in the precision of treatment effect estimates has been stretched out before (see Davies and Galla. Degree irregularity and rank probability bias in network meta-analysis, arXiv:2003.07662).

We added on page 13 the following statement citing the simulation studies by Kibret et al and by Davies and Galla: "Simulation studies using theoretical examples have shown the importance of accounting for the precision in the estimation of the treatment effects when a hierarchy is to be obtained. However, we show that cases of extreme imbalance in the precision of the treatment effects are rather uncommon."

Regarding the non-probabilistic ranking metric, the authors used a re-parametrization of the NMA model from which "the resulting hierarchy is identical to that obtained using relative effects from the conventional NMA model." I think the authors should clarify which relative effects they refer to here. For networks with a clear control intervention, it seems straightforward. But for networks without a control intervention changing the reference treatment may yield different hierarchies.

Rankings obtained by ordering the relative treatment effects (whatever the model used) do not depend on the choice of the reference treatment. By ordering the relative treatment effects the resulting hierarchy will indeed be the same if they use as a reference a specific (observed) treatment (e.g. placebo, or the worst/best treatment in the absence of a control) or an unobserved treatment assumed to have an average effect. We added an explanation in the text on page 8: "The resulting hierarchy is identical to that obtained using relative effects from the conventional NMA model, irrespective of the reference treatment."

I think the illustrative example fits better after the empirical evidence.

We discussed this and we think that for didactic reasons the example fits better before the empirical results.

It seems to me that, when investigating the influence of study features, precision of treatment effects should also be considered in relation to their magnitude. See the fictional example below where the precision for all three comparisons is the same in the two situations ($se_{BA}=.3061861$, $se_{CA}=.3061862$, $se_{CB}=.3535533$) and there is full agreement in hierarchy between relative effects (against A), SUCRA and P(best). A moderate increase in the precision of comparison CB would not affect the agreement of the right hand side ranking, whereas the same increase would affect the agreement of the ranking at the left.

We added on page 14 the following statement for clarification: "Also, the role of precision in ranking disagreement is more pronounced in cases where the interventions have similar effects."

Minor comments

The authors report that "a few networks showed a much lower agreement between the two SUCRAs (page 12). These networks provide posterior effect estimates for which the Normal approximation is not optimal." Did these networks have some specific characteristics?

We did not focus on the characteristics of these networks as they were very few so we could not explore them and generate hypotheses. However, we observed that the non-optimal Normal approximation of the posterior effect estimates were often due to rare outcomes. For example, the network with a Spearman's ρ of 0.6 between the two SUCRAs (the second lowest value observed), whose plots are reported in the supplementary figure S1, included studies with few events. We added an explanation in the text on page 12: "These networks provide posterior effect estimates for which the Normal approximation is not optimal, some of which due to rare outcomes".

Page 14, "Our results also confirm that a treatment hierarchy can differ when the uncertainty in the estimation is incorporated into the ranking metric": I am not sure I understand what the authors imply here.

We refer to the impact of the uncertainty on the treatment hierarchies (as well as on the agreement among them). For example, SUCRA encompasses the precision of the estimates so the resulting hierarchy may differ from one based on a non-probabilistic ranking metric such as the point estimate of the relative treatment effects. We added an explanation in parentheses in the text on page 14: "by using, for example, a probabilistic metric rather than ranking the point estimates of the mean treatment effects".

Page 14, "We acknowledge that there could be other factors influencing the agreement between hierarchies that we did not explore, such as the risk of bias...". I cannot see how risk of bias could influence the agreement of the ranking since none of the metrics investigated takes this aspect into account.

Thank you, you are right. We have removed "such as the risk of bias" and added the following statement on page 15 to clarify the impact of adjustment: "Adjustment via network meta-regression (for example, for risk of bias or small-study effects) might impact on the ranking of treatments not only by changing the point estimate but also by altering the total precision and the imbalance in the precision of the estimated treatment effects. We did not investigate the agreement between treatment hierarchies obtained from such adjusted analyses."

VERSION 2 – REVIEW

REVIEWER	Xiangyu MENG Team of Molecular Oncology, Institut Curie - UMR 144 - CNRS, Paris, France
REVIEW RETURNED	01-Jul-2020

GENERAL COMMENTS	Comments appropriately addressed in the revision. I would recommend publication of the work.
--

REVIEWER	Anna Chaimani Inserm, Université de Paris, France
REVIEW RETURNED	03-Jul-2020

GENERAL COMMENTS	The authors have adequately addressed all my comments. I do not have any further recommendations.
---